# Graph Agreement Models
# for Semi-Supervised Learning

**Otilia Stretcu**[‡][*], **Krishnamurthy Viswanathan**[†], **Dana Movshovitz-Attias**[†],
**Emmanouil Antonios Platanios**[‡], **Andrew Tomkins**[†], **Sujith Ravi**[†]
[†]Google Research, [‡]Carnegie Mellon University
ostretcu@cs.cmu.edu,{kvis,danama}@google.com,
e.a.platanios@cs.cmu.edu,{tomkins,sravi}@google.com

## Abstract

Graph-based algorithms are among the most successful paradigms for solving semi-supervised learning tasks. Recent work on graph convolutional networks and neural graph learning methods has successfully combined the expressiveness of neural networks with graph structures. We propose a technique that, when applied to these methods, achieves state-of-the-art results on semi-supervised learning datasets. Traditional graph-based algorithms, such as label propagation, were designed with the underlying assumption that the label of a node can be imputed from that of the neighboring nodes. However, real-world graphs are either noisy or have edges that do not correspond to label agreement. To address this, we propose *Graph Agreement Models* (GAM), which introduces an auxiliary model that predicts the probability of two nodes sharing the same label as a learned function of their features. The agreement model is used when training a node classification model by encouraging agreement only for the pairs of nodes it deems likely to have the same label, thus guiding its parameters to better local optima. The classification and agreement models are trained jointly in a co-training fashion. Moreover, GAM can also be applied to any semi-supervised classification problem, by inducing a graph whenever one is not provided. We demonstrate that our method achieves a relative improvement of up to 72% for various node classification models, and obtains state-of-the-art results on multiple established datasets.

## 1   Introduction

In many practical settings, it is often expensive, if not impossible, to obtain large amounts of labeled data. Unlabeled data, on the other hand, is often readily available. Semi-supervised learning (SSL) algorithms leverage the information contained in both the labeled and unlabeled samples, thus often achieving better generalization capabilities than supervised learning algorithms. Graph-based semi-supervised learning [43, 41] has been one of the most successful paradigms for solving SSL problems when a graph connecting the samples is available. In this paradigm, both labeled and unlabeled samples are represented as nodes in a graph. The edges of the graph can arise naturally (e.g., links connecting Wikipedia pages, or citations between research papers), but oftentimes they are constructed automatically using an appropriately chosen similarity metric. This similarity score may also be used as a weight for each constructed edge (e.g., for a document classification problem, Zhu et al. [43] set the edge weights to the cosine similarity between the tf-idf vectors of documents).

There exist several lines of work that leverage graph structure in different ways, from label propagation methods [43, 41] to neural graph learning methods [7, 37] to graph convolution approaches [15, 35], which we describe in more detail in Sections 2 and 5. Most of these methods rely on the assumption that graph edges correspond in some way to label similarity (or *agreement*). For instance, label propagation assumes that node labels are distributed according to a jointly Gaussian distribution

---

[*]This work was done during an internship at Google.

whose precision matrix is defined by the edge weights. However, in practice, graph edges and their weights come from noisy sources (especially when the graph is constructed from embeddings). Therefore, the edges may not clearly correspond to label agreement uniformly across the graph. The likelihood of two nodes sharing a label could perhaps be better modeled explicitly, as a learned function of their features. To this end, we introduce *graph agreement models* (GAM), which learn to predict the probability that pairs of nodes share the same label. In addition to the main node classification model, we introduce an auxiliary agreement model that takes as input the feature vectors of two graph nodes and predicts the probability that they have the same label. The output of the agreement model can be used to regularize the classification model by encouraging the label predictions for two nodes to be similar only when the agreement model says so. Intuitively, a perfect agreement model will allow labels to propagate only across "correct" edges and will thus make it possible to boost classification performance using noisy graphs.

Training either the classification or the agreement model in isolation may be hard, if not impossible, for many SSL settings. That is because we often start with a tiny number of labeled nodes, but a large number of unlabeled nodes. For example, the agreement model needs to be supervised with edges connecting labeled nodes, and in some cases, due to the scarcity of labeled data, there may not be any such edges to begin with. To ameliorate this issue, we also propose a learning algorithm that allows the classification and agreement models to be jointly trained. While the agreement model can be used to regularize the classification model, the most confident predictions of the

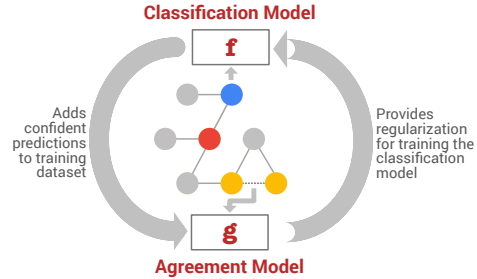

Figure 1: Proposed learning paradigm.

latter can be used to augment the training data for the agreement model. Figure 1 illustrates the interaction between the two models. This idea is inspired by the *co-training* algorithm proposed by Blum and Mitchell [6]. We show in our experiments that the proposed approach achieves the best known results on several established graph-based classification datasets. We also demonstrate that our approach works well with graph convolutional networks [15], and the combination outperforms graph attention networks [35] which are expensive during inference.

While our method was originally inspired by graph-based classification problems, we show that it can also be applied to *any semi-supervised classification problem*, by *inducing a graph* whenever one is not provided. We performed experiments on the popular datasets CIFAR-10 [16] and SVHN [26] and show that GAM outperforms state-of-the-art SSL approaches. Furthemore, the proposed method has the following desirable properties:

1. **General:** Can be applied on top of any classification model to improve its performance.
2. **State-of-the-Art:** Outperforms previous methods on several established datasets.
3. **Efficient:** Does not incur any additional performance cost at inference.
4. **Robust:** Up to $18\%$ accuracy improvement when $5\%$ of the graph edges correspond to agreement.

## 2 Background

We introduce notation used in the paper, and describe related work most relevant to our proposed method. Let $\mathcal{G}(V, E, W)$ be a graph with nodes $V$, edges $E$, and edge weights $W = \{w_{ij}\}_{ij \in E}$. Each node $i \in V$ is represented by a feature vector $x_i$ and label $y_i$. Labels are observed for a small subset of nodes, $L \subset V$, and the goal is to infer them for the remaining unlabeled nodes, $U = V \setminus L$.

Graph-based algorithms, such as label propagation (LP), tackle this problem by assuming that two nodes connected by an edge likely have the same label, and a higher edge weight indicates a higher likelihood that this is true. In LP, this is done by encouraging a node's predicted label probability distribution to be equal to a weighted average of its neighbors' distributions. While this method is simple and scalable, it is limited as it does not take advantage of the node features. Weston et al. [37] and Bui et al. [7] propose combining the LP approach with the power of neural networks by learning expressive node representations. In particular, Bui et al. [7] propose *Neural Graph Machines (NGM)*, a method for training a neural network that predicts node labels solely based on node features and the LP assumption which takes the form of regularization. They minimize the following objective:

$$\mathcal{L}_{\text{NGM}} = \underbrace{\sum_{i \in L} \ell(f(x_i), y_i)}_{\text{supervised}} + \lambda_{LL} \underbrace{\sum_{i,j \in L, ij \in E} w_{ij} d(h_i, h_j)}_{\text{labeled-labeled}} + \lambda_{LU} \underbrace{\sum_{i \in L, j \in U, ij \in E} w_{ij} d(h_i, h_j)}_{\text{labeled-unlabeled}} + \lambda_{UU} \underbrace{\sum_{i \in U, j \in U, ij \in E} w_{ij} d(h_i, h_j)}_{\text{unlabeled-unlabeled}},$$

where $f(x_i)$ is the predicted label distribution for node $i$, $h_i$ is the last hidden layer representation of the network for input $x_i$, $\ell$ is a cost function (e.g., cross-entropy), and $d$ is a loss function that measures dissimilarity between representations (e.g., L2). $\lambda_{LL}$, $\lambda_{LU}$, and $\lambda_{UU}$ are positive constants representing regularization weights applied to distances between node representations for edges connecting two labeled nodes, a labeled and an unlabeled node, and two unlabeled nodes, respectively. Intuitively, this objective function aims to match predictions with labels, for nodes where labels are available, while also making node representations similar for neighboring nodes in the graph.

NGMs are used to train neural networks and learn complex node representations, they are scalable, and they incur no added cost at inference time as the classification model $f$ is unchanged. However, the quality of learned parameters relies on the quality of the underlying graph. Most real-world graphs contain spurious edges that may not directly reflect label similarity. In practice, graphs are of one of two types: (1) Provided: As an example, several benchmark datasets for graph-based SSL consider a citation graph between research articles, and the goal is to classify the article topic. While articles with similar topics often cite each other, there exist citations edges between articles of different topics. Thus, this graph offers a good but non-deterministic *prior* for label agreement. (2) Constructed: In many settings, graphs are not available, but can be generated. For example, one can generate graph edges by calculating the pairwise distances between research articles, using any distance metric. The quality of this graph then depends on how well the distance metric reflects label agreement.

In either case, edge weights may not correspond to likelihood of label agreement, and given a small number of labeled nodes, it is hard to determine whether that correspondence exists in a given graph. This drastically limits the regularization capacity of label propagation methods: a large regularization weight risks disrupting the base model due to noisy edges, while a small regularization weight does not prevent the base model from overfitting. In the next section, we propose a novel approach that aims to address this problem, and can be thought of as a generalization of label propagation methods.

## 3 Proposed Method

We propose *Graph Agreement Models (GAM)*, a novel approach that aims to resolve the main limitation of label propagation methods while leveraging their strengths. Instead of using the edge weights as a fixed measure of how much the labels of two nodes should agree, GAM learns the probability of agreement. To achieve this, we introduce an *agreement model*, $g$, that takes as input the features of two nodes and (optionally) the weight of the edge between them, and predicts the probability that they have the same label. The predicted agreement probabilities are then used when training the *classification model*, $f$, to prevent overfitting.

**Classification Model.** The only aspect of the classification model that we modify is the loss function. We propose a modified version of the NGM loss function, where the weight of each edge's contribution to the loss is decided by the agreement model. In other words, we replace all $w_{ij}$s with $g_{ij} = g(x_i, x_j, w_{ij})$. The new loss function becomes:

$$\mathcal{L}_{\text{GAM}} = \sum_{i \in L} \ell(f_i, y_i) + \lambda_{LL} \sum_{i,j \in L, ij \in E} g_{ij} d(y_i, f_j) + \lambda_{LU} \sum_{i \in L, j \in U, ij \in E} g_{ij} d(y_i, f_j) + \lambda_{UU} \sum_{i,j \in U, ij \in E} g_{ij} d(f_i, f_j), \quad (1)$$

where we use the short notation $f_i = f(x_i)$. Note that there are actually a few more differences between $\mathcal{L}_{\text{NGM}}$ and $\mathcal{L}_{\text{GAM}}$. Since the agreement model $g$ is designed to estimate agreement between labels, and not between the hidden representations $h$ generated by $f$, we are in fact penalizing disagreement between the predicted label distributions directly. This is also easier to implement for arbitrary classification models, since it removes the need for a decision on what should be the hidden representation of the graph nodes. Moreover, our regularization terms make use of the supervised node labels, whenever available (i.e., in the LU term, or one of the two sides of the LL term). This is because we aim to decrease the entropy of the predictions, which, as we have empirically observed, improves the stability of the learning process.

**Agreement Model.** The agreement model, $g$, can be any neural network. The only constraint is that it must receive the features of two nodes and predict a single value that represents the probability that the two nodes have the same label. Note that using the edge weight could be helpful, but is not necessary. Since modularity enables more flexibility, we decided to split the agreement model further into the following components:

1. `Encoder`: Produces a vector representation for a node. The same encoder network is applied to both inputs (each input is a node's features) of the agreement model.
2. `Aggregator`: Combines the encoded representations of the two node arguments into a single vector, and is invariant to the order of its two arguments (e.g., the "sum" operation). The last

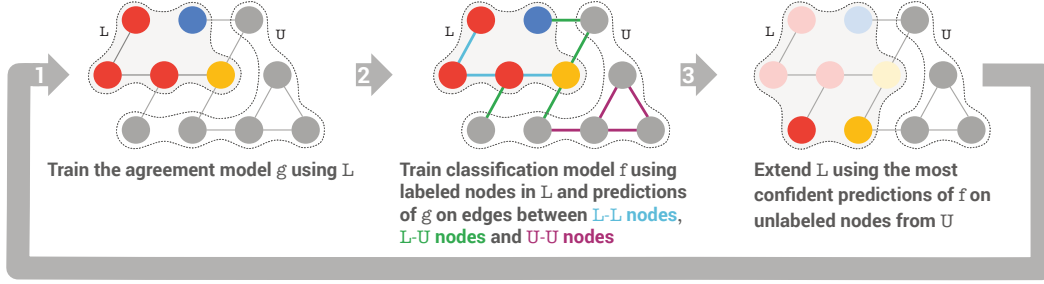

Figure 3: Overview of the three main steps in each iteration of the proposed co-training algorithm.

condition represents a meaningful and valid inductive bias for the agreement model, namely that the order in which nodes are presented should not influence their probability of agreement.

3. `Predictor`: Given the aggregator output, this component predicts the probability that the initial two nodes have the same label.

Figure 2 shows how these components are used together in the agreement model. This formulation is highly generic, as each module can be implemented as an arbitrary neural network. The recent success of BERT [10]—a transfer learning architecture that recently achieved state-of-the-art performance for several natural language processing tasks—seems to indicate that it is important to have a highly expressive `encoder`, even if the `predictor` is only a linear function. Furthermore, it is clear that the choice for an `encoder` network heavily depends on the nature of the data (e.g., convolutional neural networks perform well for images). However, through our extensive experiments—which are further described in Section 4—we observed that simple multi-layer perceptrons consistently provide a good trade-off between performance and efficiency. Regarding the `aggregator`, "addition" and "subtraction" are both simple and valid options. However, the functional form that seemed to work best in practice, and the one we use in our experiments is defined as $\texttt{aggregator}(e_i, e_j) = (e_i - e_j)^2$, where $e_i$ and $e_j$ are the output vector embeddings from the `encoder` for nodes $i$ and

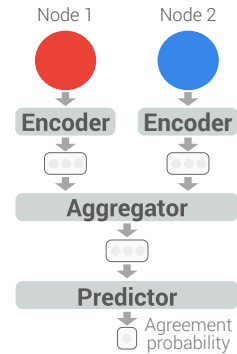

Figure 2: Agreement model components.

$j$, respectively. This function is invariant to the order of the two nodes and it reflects our intuition that agreement probability can be thought of as distance between two nodes in a latent space. For the `predictor`, we use a linear layer, similar to BERT. Finally, we use the following loss function to train the agreement model:

$$\mathcal{L}_{\text{GAM}}^{\text{agreement}} = \sum_{i \in L, j \in L, ij \in E} \ell(g(x_i, x_j, w_{ij}), \mathbb{1}_{y_i = y_j}), \tag{2}$$

where $\ell$ is a binary classification loss function (e.g., sigmoid cross-entropy), and $\mathbb{1}_{y_i=y_j}$ is an indicator function whose value is 1 when $y_i = y_j$, and 0 otherwise. It now remains to describe the overall learning algorithm we propose for jointly training the classification and agreement models.

## 3.1 Learning Algorithm

The classification model, $f$, is trained by minimizing the loss function shown in Equation 1. However, this loss function uses the agreement model $g$, that also needs to be trained. We can think of $g$ as regularizing the training process of $f$. Perhaps most interestingly though, while the agreement model can play a crucial role in training the classification model, the classification model can also help train the agreement model. A key contribution of our work is exactly this interaction in the training processes of $f$ and $g$. More specifically, we propose the following learning algorithm:

1. Train the agreement model $g$ to convergence, using the limited amount of labeled data that is provided. We refer to the initial trained model as $g^0$.
2. Train $f$ using $g^0$ in its loss function. We refer to the trained model as $f^0$.
3. Let $f^0$ produce predictions over all of the unlabeled nodes. Although this model was trained using a limited amount of data, we expect its most confident predictions (i.e., the labels with the highest probability) to most likely be correct. Thus, we take the top $M$ most confident predictions and add them to the set of labeled nodes. We refer to this step as the *self-labeling* phase.

The newly added labeled examples can provide new information to the agreement and classification models. We thus start again from step 1, and obtain new trained models $g^1$ and $f^1$, and a new set of

most confident $M$ predictions for the remaining unlabeled nodes. We repeat this process for $k$ steps (or until all nodes have been labeled), using $g^{k-1}$ to help train $f^k$ and $f^k$ to help train $g^{k+1}$.

This training algorithm resembles the co-training algorithm, originally proposed by Blum and Mitchell [6]. The core idea behind it is that, if $f$ and $g$ are good at exploiting different kinds of information, then we can leverage that by having them help train each other. Similar algorithms have been successfully used in practice [e.g., 22], and, for some settings there even exist theoretical guarantees that such algorithms will converge to a better classifier than the one that would have been obtained without co-training [3]. For these reasons, we expect this interaction to boost the performance of both $f$ and $g$. An illustration of the algorithm is shown in Figure 3.

Note that $g$ only participates in training. At inference time predictions are made by applying the trained $f$ to the input. Thus GAM does not incur extra computation cost at inference.

### 3.2 Inducing Graphs

Methods that rely on the provided graph have two main limitations. First, they cannot be applied to datasets that do not include a graph. In addition, by inspecting Equation 1 it is easy to notice that even with $g$ providing perfect predictions, it will only allow labels to propagate along the graph edges connecting nodes with matching labels. However, if the graph is sparse or the number of labeled nodes is small, there may be unlabeled nodes for which there is no "agreement" path connecting it to a labeled node from its class. In fact, in the benchmark datasets, Cora [19], Citeseer [5] and Pubmed [25], propagating labels through "agreement" edges, while starting at the provided labeled nodes, only covers 84%, 49%, and 85% of the nodes respectively. The remaining nodes do not appear in any of the regularization terms of the classification model loss function, thus making it prone to overfitting. Our approach alleviates this issue by self-labeling unlabeled nodes. These nodes can then propagate their labels during the next co-training iteration.

In fact, we propose to go a step further and address both limitations. Notice that $g$ can be trained and applied on any pair of labeled nodes—not necessarily connected by an edge—and can thus regularize predictions made by $f$ for any pair of nodes. This can be achieved by removing all constraints $ij \in E$ from Equations 1 and 2. In this formulation the provided graph becomes unnecessary. This is equivalent to having a fully-connected graph, and using the agreement model to denoise it. We refer to this GAM variant that does not use a graph as **GAM\***.

Our experimental results, presented in the next section, indicate that this new formulation not only boosts the performance of GAM on some graph-based datasets, but it also opens up a wide range of new applications. That is because GAM\* can now be applied to any SSL dataset, whether or not a graph is provided. In Section 4.2, we evaluate GAM\* on two datasets with no inherent graph structure, and show that it is able to improve upon state-of-the-art methods for SSL.

## 4 Experiments

We performed a set of experiments to test different properties of GAM. First, we tested the **generality** of GAM by applying our approach to Multilayer Perceptrons (MLP), Convolutional Neural Networks (CNN), Graph Convolution Networks (GCN) [15], and Graph Attention Networks (GAT) [35][2]. Next, we tested the **robustness** of GAM when faced with noisy graphs, as well as evaluated GAM and GAM\* **with and without a provided graph**, comparing them with the state-of-the-art methods.

### 4.1 Graph-based Classification

**Datasets.** We obtained three public datasets from Yang et al. [38]: Cora [19], Citeseer [5], and Pubmed [25], which have become the de facto standard for evaluating graph node classification algorithms. We used the same train/validation/test splits as Yang et al. [39], which have been used by the methods we compare to. In these datasets, graph nodes represent research publications and edges represent citations. Each node is represented as a vector, whose components correspond to words. For Cora and Citeseer the vector elements are binary indicating whether the corresponding term is present in the publication, while for Pubmed they are real-valued tf-idf scores. The goal is to classify research publications according to their main topic which belongs to a provided set of topics. In each case we are given true labels for a small subset of nodes. Dataset statistics are shown in Table 4 in Appendix A.

**Setup.** We implemented our models in TensorFlow [1]. Parameter updates are using the Adam optimizer [14] with default TensorFlow parameters, and initial learning rate of 0.001 for MLPs and GCN, and 0.005 for GAT (based on the original publication [35]). When training the classification model, we used a batch size of 128 for both the supervised term and for the edges in each of the $LL$, $LU$, and $UU$ terms. We stopped training when validation accuracy did not increase in the last 2000 iterations, and reported the test accuracy at the iteration with the best validation performance. For the agreement model, we sampled random batches containing pairs of nodes from the pool of all edges with both nodes labeled for GAM, or of all pairs of nodes for GAM*. In both cases, we ensured a ratio of 50% positives (labels agree) and 50% negatives (labels disagree). In the case of GAM, since graphs typically contain more positive edges than negative, extra negative samples were selected at random from the pairs of nodes with no edge connecting them. Our experiments were performed using a single Nvidia Titan X GPU, and our implementation can be found at `https://github.com/tensorflow/neural-structured-learning`.

**Models.** For both GAM and NGM, we used Euclidean distance for $d$, and we selected $\lambda_{LL}$, $\lambda_{LU}$, and $\lambda_{UU}$ based on validation set accuracy, where we varied $\lambda_{LU} \in \{0.1, 1, 10, 100, 1000, 10000\}$, and set $\lambda_{UU} = \frac{1}{2}\lambda_{LU}$ and $\lambda_{LL} = 0$ (we found through experimentation that the $LL$ component does not have a significant contribution, probably because the predictions for labeled nodes are already accounted for in the supervised loss term). For the agreement model, we used an MLP with the same number of hidden units as the classification model. We started with 20 labeled examples per class and, when extending the labeled node set, we added the $M$ most confident predictions of the classifier over unlabeled nodes. In our experiments, we set $M = 200$, but doing parameter selection for $M$ as well could potentially lead to even better results. To avoid adding incorrectly-labeled nodes we filtered out predictions where the classification confidence (i.e., the maximum probability assigned to one of the labels) was lower than 0.4 (since the smallest number of classes considered is 3 for Pubmed, making chance classification probability 0.33).

Table 1: Test classification accuracies (%) on graph-based datasets. The first section contains results reported in related work. The next segments show results for different classifiers and their extensions using NGM, VAT, GAM, and GAM*. Subscripts refer to the number of hidden units. Shaded methods do not use the graph.

| Model | Datasets | | |
|---|---|---|---|
| | Cora | Citeseer | Pubmed |
| ManiReg [4] | 59.5 | 60.1 | 70.7 |
| SemiEmb [37] | 59.0 | 59.6 | 71.7 |
| LP [43] | 68.0 | 45.3 | 63.0 |
| DeepWalk [28] | 67.2 | 43.2 | 65.3 |
| ICA [19] | 75.1 | 69.1 | 73.9 |
| Planetoid [39] | 75.7 | 64.7 | 77.2 |
| Chebyshev [8] | 81.2 | 69.8 | 74.4 |
| $\text{MLP}_{[250, 100]}$+NGM [7] | – | – | 75.9 |
| MoNet [24] | 81.7 | – | 78.8 |
| $\text{GCN}_{16}$ [15] | 81.5 | 70.3 | 79.0 |
| $\text{GAT}_8$ [35] | 83.0 | 72.5 | 79.0 |
| $\text{GCN}_{16}$ + O-BVAT [9] | 83.6 | 74.0 | 79.9 |
| $\text{MLP}_{128}$ | 51.7 | 52.2 | 69.4 |
| $\text{MLP}_{128}$ + NGM | 77.7 | 67.8 | 73.6 |
| $\text{MLP}_{128}$ + VAT | 56.5 | 56.1 | 73.1 |
| $\text{MLP}_{128}$ + VATENT | 24.1 | 46.7 | 70.1 |
| $\text{MLP}_{128}$ + GAM | **80.7** | **73.0** | **82.8** |
| $\text{MLP}_{128}$ + GAM* | 70.7 | 70.3 | 71.9 |
| $\text{MLP}_{4 \times 32}$ | 46.6 | 49.0 | 68.7 |
| $\text{MLP}_{4 \times 32}$ + NGM | 77.6 | 63.1 | 70.2 |
| $\text{MLP}_{4 \times 32}$ + VAT | 55.3 | 46.5 | 74.2 |
| $\text{MLP}_{4 \times 32}$ + VATENT | 33.0 | 29.1 | 62.5 |
| $\text{MLP}_{4 \times 32}$ + GAM | **80.1** | **70.4** | **79.3** |
| $\text{MLP}_{4 \times 32}$ + GAM* | 64.0 | 66.9 | 76.9 |
| $\text{GCN}_{128}$ | 80.9 | 68.1 | 76.9 |
| $\text{GCN}_{128}$ + NGM | 81.4 | 68.9 | 76.2 |
| $\text{GCN}_{128}$ + VAT | 79.0 | 69.5 | 76.8 |
| $\text{GCN}_{128}$ + VATENT | 83.4 | 69.8 | 75.0 |
| $\text{GCN}_{128}$ + GAM | **86.2** | **73.5** | **86.0** |
| $\text{GCN}_{128}$ + GAM* | 84.2 | 71.3 | 77.0 |
| $\text{GCN}_{1024}$ | 81.3 | 70.5 | 78.5 |
| $\text{GCN}_{1024}$ + NGM | 82.0 | 70.5 | 68.9 |
| $\text{GCN}_{1024}$ + VAT | 81.8 | 69.3 | 76.3 |
| $\text{GCN}_{1024}$ + VATENT | 64.0 | 50.5 | 72.1 |
| $\text{GCN}_{1024}$ + GAM | **86.0** | **73.6** | **81.6** |
| $\text{GCN}_{1024}$ + GAM* | 82.4 | 71.9 | 81.2 |
| $\text{GAT}_{128}$ | 81.6 | 69.0 | – |
| $\text{GAT}_{128}$ + NGM | 80.3 | 70.8 | – |
| $\text{GAT}_{128}$ + GAM | 84.3 | 70.3 | – |
| $\text{GAT}_{128}$ + GAM* | **85.0** | **73.6** | – |

**Results.** Our results are reported in Table 1. Results obtained with GAM are denoted in the form "{base model} + GAM". The subscript following the base model represents the number of hidden units of the classification model (e.g., $\text{MLP}_{128}$ is a multilayer perceptron with a single layer of 128 hidden units, and $\text{MLP}_{4 \times 32}$ is a multilayer peceptron with 4 layers of 32 hidden units each). We also report the best known results for these datasets from other publications, as reported in [35]. Furthermore, in order to allow for a more complete comparison with other general-purpose SSL methods, we also compared with VAT [23]—the current state-of-the-art SSL method, as reported in [27] and [23]—and its entropy minimization variant, VATENT. We set the VAT regularization weight to 1, as in [23, 27]. The results can be summarized as follows:

- GAM always improves the classification model accuracy of the base model, for all base models, often by a significant margin (e.g., $+33.5\%$ for $MLP_{4 \times 32}$ on Cora, which is a relative increase of $72\%$). Note that we measure relative performance as $\frac{\text{new\_accuracy} - \text{baseline\_accuracy}}{\text{baseline\_accuracy}}$.
- GAM also consistently achieves important gains compared to NGM (e.g., $+4.8\%$ for $GCN_{128}$ on Cora, and $+9.8\%$ on Pubmed), supporting the intuition behind our edge denoising approach.
- VAT also consistently improves upon the baseline classifier (although not as much as GAM or GAM*), even though it does not use the graph and it treats the unlabeled nodes as independent samples. Interestingly, VATENT fails on these datasets in many cases, although the same method performs very well on other SSL datasets (Section 4.2).
- It is interesting to note that although GCN and GAT already use the graph as part of their architecture, their performance can be further improved by using GAM.
- To the best of our knowledge, the GAM variants obtain the best results reported on these datasets.
- Further, note that GCN with GAM outperforms GAT (which is GCN with attention), suggesting that GAM regularization is a better alternative to attention for handling noisy graphs. Note that the GAT results for Pubmed are missing because we use the implementation of GAT provided by Veličković et al. [35], and it runs out of GPU memory for 128 hidden units on Pubmed.

**Robustness.** We developed GAM with the goal of being able to handle graphs with "incorrect" edges (i.e. those that connect nodes with differing labels). We consider such edges "incorrect" under the label propagation assumption, despite the fact that they may refer to real-world connections between these nodes (e.g., citations between research articles on different topics). In Cora, Citeseer, and Pubmed, 19%, 26%, and 20% of the edges, respectively, are incorrect. To demonstrate the ability of GAM to handle these incorrect edges and perhaps even higher levels of noise, we performed a robustness analysis by introducing spurious edges to the graph, and testing whether our agreement model learns to ignore them. We added spuri-

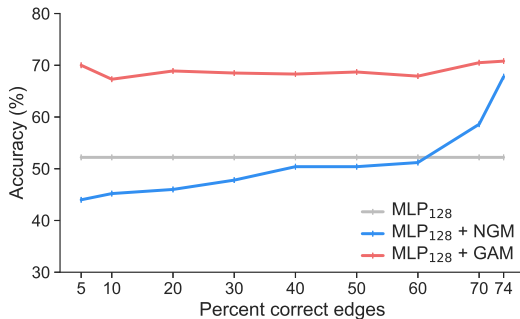

Figure 4: Robustness to noisy graphs. The x axis represents the percentage of correct edges remaining after adding wrong edges to the Citeseer dataset.

ous edges by randomly sampling pairs of nodes with different true labels until the percentage of incorrect edges met a desired target. We tested the performance of GAM on a set of graphs created in this manner. MLPs are good base model candidates for testing this because they can only be affected by the graph quality through the GAM regularization terms (unlike GCN or GAT, where the graph is implicitly used in the model). The results are shown in Figure 4 on the Citeseer dataset (the hardest of the three datasets), for graphs containing between $5\%$ and $74\%$ correct edges. A plain MLP with 128 hidden units obtains $52.2\%$ accuracy independent of the level of noise in the graph. Adding GAM to this MLP increases its accuracy by about $19\%$. This improvement persists even as the fraction of correct edges decreases. For example, the accuracy remains $70\%$ even in the case where only $5\%$ of the graph edges are correct. In contrast, the performance of NGM steadily decreases as the fraction of incorrect edges increases, to the point where it starts performing worse than the plain MLP (when the percent of correct edges $\leq 60\%$), and it is thus preferable not to use it.

**Ablation Study.** We performed experiments to show how much each component of GAM contributes to its success, as follows:

**(1)** Perfect agreement: We evaluated how well GAM would perform if the agreement model produced perfect predictions. This is done by letting the agreement model see the true labels, and always return 1 when nodes agree, and 0 otherwise. We ran this experiment for all 3 datasets with an MLP base classifier. The results in Table 2 show that a perfect agreement model produces a huge boost, up to 38.8% over the baseline. For Citeseer, the smaller improvement is not surprising, given that only 49% nodes are connected by agreement (see Section 3.2).

Table 2: Accuracy (%) of an MLP with 128 hidden units using GAM with a perfect agreement model.

| Model | Datasets | | |
|---|---|---|---|
| | Cora | Citeseer | Pubmed |
| $MLP_{128} + GAM_p$ | 90.5 | 76.5 | 91.6 |

**(2)** Sensitivity to agreement model: We evaluated how sensitive GAM is to the choice of agreement model architecture. To assess this, we ran GAM multiple times, with a fixed classification model architecture and we various agreement model sizes. Figure 6 in Appendix C shows the test accuracy

per co-train iteration for each of these models. The results indicate that the behaviour of GAM is stable with respect to the agreement model size, which suggests that the agreement model size is a hyperparameter that does not require much tuning effort.

**(3)** Self-labeling: We evaluated the usefulness of the self-labeling component by showing how the test accuracy evolves after each co-training iteration. Figure 5 shows that the accuracy generally has an increasing trend with more co-training iterations. In some cases, the final iterations may have a decreasing trend, because in the last few iterations the model self-labels the samples that it is most uncertain about, and thus it is more likely to make mistakes. For this reason, we kept track of the validation accuracy, and at the end we restored the model from the co-train iteration with the best validation accuracy. Self-labeling is also a critical component for datasets such as Pubmed, where in the first co-train iteration there are no edges with both nodes labeled, so $g$ cannot be trained until we self-label more nodes. In such cases, $g$ returns 1 by default until it can be trained, defaulting to NGM and relying on the graph (although for noisy graphs, one could return 0 by default).

## 4.2 Semi-Supervised Learning Without a Graph

Our robustness experiments show that GAM is effective even when the majority of edges in the graph connect nodes with mismatched labels. Therefore, we tested its power further by considering a more extreme scenario: no graph is provided, and the agreement model is tasked with learning whether an arbitrary pair of nodes shares a label. Note that having no graph, and picking random pairs of samples to use in the regularization terms in Equation 1, is equivalent to having a fully-connected graph from which we sample edges. We tested this scenario on Cora, Citeseer, and Pubmed and the results are marked as GAM* in Table 1. For completeness, we also show results for GCN+GAM* and GAT+GAM*, where even though the GAM* regularization term does not use the graph, the classification models use it by design. Our results show that GAM* also boosts the performance of all tested baseline models, with a gain of up to $19\%$ accuracy for MLPs, $3.3\%$ for GCNs, and $4.6\%$ for GATs. It is worth noting that, even though GAM outperforms GAM* due to the extra information provided by the graph, GAM* generally outperforms the competing methods that also do not use a graph, and often even NGM which does.

**Non-graph Datasets.** Since our approach no longer requires a graph to be provided, we tested GAM on the popular CIFAR-10 [16] and SVHN [26] datasets. For evaluation, we use the setup and train/validation/test splits provided by [27], which aims to provide a realistic framework for evaluating SSL methods. Thus, we start with 4000 and 1000 labeled samples for CIFAR-10 and SVHN, respectively, while the remaining training samples are considered unlabeled. More information about these datasets can be found in Appendix B. It is important to note that while Cora, Citeseer and Pubmed were evaluated under a **transductive** setting (where the input features and the graph structure of the test nodes are seen during training, but not their labels) as is typical in graph-based SSL, in the following experiments we evaluate GAM* under an **inductive** setting (the features of the test nodes are completely held out, and there is no graph to provide other information about them).

**Models.** As the datasets consist of images, we use a Convolutional Neural Network (CNN) with 2 convolution layers followed by max-pooling, then 2 fully-connected layers (architecture details in Appendix D). The agreement model is a 3 layer MLP with 128, 64, and 32 hidden units, respectively, and Leaky ReLU activations. After each co-training iteration, we self-label 1000 unlabeled samples, subject to a confidence $> 0.4$ (same as in Section 4.1, but tuning may improve results further). VAT and VATENT settings are the same as in Section 4.1.

Table 3: Classification accuracies (%) on CIFAR-10 with 4000 labels, and SVHN with 1000 labels.

| Model | Datasets | |
|---|---|---|
| | CIFAR-10 | SVHN |
| CNN | 62.57 | 72.33 |
| CNN + VAT | 64.37 | 70.26 |
| CNN + VATENT | 66.73 | 81.86 |
| CNN + GAM* | 69.27 | 83.43 |
| CNN + VAT + GAM* | **69.64** | **85.47** |
| CNN + VATENT + GAM* | 67.29 | 84.63 |

**Results.** Table 3 shows that GAM* significantly improves performance over the baseline classifier, even when no graph is given (up to 13% on SVHN). Moreover, it can improve performance over one of the best current SSL methods, VAT, when applied in conjunction with it (e.g., +5.27% when GAM* is applied on top of VAT on CIFAR-10, which yields a 7% improvement from a plain CNN). We show the progression of the test accuracy per co-train iteration on CIFAR-10 in Figure 5 (b). Moreover, we did not tune the parameters of the CNN or the learning rate to be favorable to our method. However, the results indicate that GAM* offers a promising direction for general-purpose SSL.

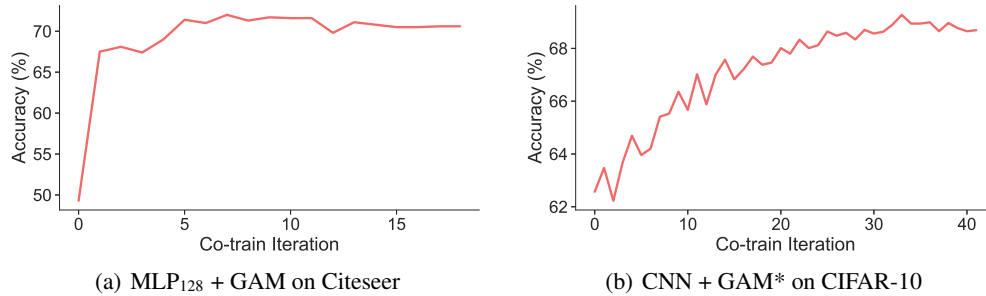

(a) MLP$_{128}$ + GAM on Citeseer       (b) CNN + GAM* on CIFAR-10

Figure 5: Test accuracy per co-train iteration for a **(a)** MLP$_{128}$ + GAM on the Citeseer dataset starting with 120 labeled samples, and self-labeling 200 samples per co-train iteration, and **(b)** CNN + GAM* on the CIFAR-10 dataset starting with 4000 labeled samples, and self-labeling 1000 samples per co-train iteration. Iteration 0 shows the baseline model accuracy, without GAM.

## 5 Related Work

There has been substantial work on graph-based semi-supervised learning [e.g., 34]. A first class of methods regularize the predicted labels using the Laplacian of the graph without taking advantage of the node features. These include label propagation [43, 41], manifold regularization [4], and ICA [19]. Another line of work [18, 20] focuses on refining the SSL graphs obtained from similarity matrices using only the similarity scores, but ignoring the node features. Recent approaches have attempted to marry the core idea behind these methods with the expressive power afforded by neural networks. Among these, the regularization based approaches of Weston et al. [36], Weston et al. [37], as well as Neural Graph Machines by [7] (described in Section 2) are closest to ours. Moreover, Planetoid [39] applies regularization using a term that depends on the skip-gram representation of the graph. Note that the notion of using agreement in predictions made by classifiers is a concept that has also been used more broadly in the context of SSL [e.g., 29], and not just for graph-based SSL. Another class of techniques learns node embeddings that take into account both the features and the graph, which are then consumed by standard supervised learning methods [28, 13, 31, 11]. More recently, there has been a large amount of work on Graph Neural Networks that extend neural networks to graph-structured inputs. See for example [42] for a survey of methods in this category. Among these, the most relevant to our work are graph convolutional networks (GCN) proposed by Kipf and Welling [15] and a scalable extension [40]. These approaches define a notion of graph convolution and uses an approximation of the convolution to provide a scalable method that produced state-of-the-art results. Moreover, [35] and [33] applied attention on the edges of the graph to further improve the performance of GCN.

Aside from graph-based approaches, there has also been a great deal of work on SSL methods without a graph. Most relevant to our work are methods that use regularization to discourage the model from making vastly dissimilar predictions for similar inputs. These include Π-Model [17, 30], Mean Teacher [32] and Virtual Adversarial Training (VAT) [23], SNTG [21] and fast-SWA [2]. Some of the best results are obtained by combining VAT with entropy minimization [12], which adds a loss term that encourages more confident predictions. SNTG infers a similarity graph between samples, but it does so in a significantly different way than GAM*. Also, in contrast to SNTG, we propose an additional self-training component, and our method is applicable when a graph is provided, whereas SNTG, as published, is not designed to use information from a provided graph.

Our proposed method, GAM, can be applied as an extension to all of the above methods, as it only requires the addition of a regularization term to their loss function.

## 6 Conclusions

We introduced Graph Agreement Models (GAM), a novel regularization method for graph-based and general purpose semi-supervised learning (SSL), that can be applied on top of any classification model. The key idea behind our approach is the interaction between a node classification model and a node agreement model, which are trained in tandem in a co-training fashion. Our experiments show that GAM can improve the accuracy of several types of classifiers, including two of the most successful graph-based SSL methods, thus establishing a new state-of-the-art for graph-based classification. Moreover, we demonstrated that GAM can be extended to settings where a graph is not provided, and it is able to improve upon the performances of some of the best SSL classification models.

## Footnotes

[2]MLPs and CNNs are common in many SSL problems and GCN and GAT achieve state-of-the-art performance on three datasets commonly used in recent graph-based SSL work.

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
