[Supplementary Material · appendix.pdf]

# Appendix

## A    Citation Networks Dataset Statistics

Table 4: Statistics for the commonly used citation networks datasets.

| DATASET | # NODES | # EDGES | # CLASSES |
|---------|---------|---------|-----------|
| CORA | 2,708 | 5,429 | 7 |
| CITESEER | 3,327 | 4,732 | 6 |
| PUBMED | 19,717 | 44,338 | 3 |

## B    Semi-Supervised Learning Datasets with No Graph

CIFAR-10 consists of $32 \times 32$ color images belonging to $10$ classes. The preprocessed dataset obtained from Oliver et al. [27] contains $45,000$ training samples, $5000$ validation samples, and $10,000$ test samples. To convert this to a SSL setting, we only allow our model to see $4000$ labeled samples (as selected by Oliver et al. [27]), while for the rest of the training samples our model only has access to their features, but not their true labels.

Similarly, the SVHN (Street View House Numbers) dataset consists of $32 \times 32$ color images from $10$ classes, coming from real world images depicting house numbers from Google Street View images[3]. Using the splits from Oliver et al. [27], the dataset contains $65,931$ training samples, $7,326$ validation samples, and $26,032$ test samples. We only allow our model to see the labels for $1,000$ training samples.

For CIFAR-10, we perform global contrast normalization and ZCA whitening, and for SVHN we simply convert the pixels to [-1, 1] range, as done by Oliver et al. [27].

## C    Sensitivity to Agreement Model Architecture

Figure 6: Test accuracy per co-train iteration with GAM for various agreement model sizes, on the Citeseer dataset. The classification model size for all plots is a multilayer perceptron with 128 hidden units. We vary the number of agreement model hidden units between $8$ to $2048$. Iteration 0 shows the baseline model accuracy, without GAM.

## D    Convolutional Neural Network Architecture

The architecture we chose was inspired by the TensorFlow CIFAR tutorial found at `https://github.com/tensorflow/models/blob/adc271722e512c8a55ef18b46c666a031de77774/tutorials/image/cifar10/cifar10.py`, including numbers of hidden units and weight decay values.

In summary, the CNN contains the following layers:

1. Convolution: 2D convolution using a $5 \times 5$ filter with 64 channels, followed by ReLU activation.

2. Max pooling using a $3 \times 3$ window.

3. Local Response Normalization.

4. Convolution: 2D convolution using a $5 \times 5$ filter with 64 channels, followed by ReLU activation.

5. Local Response Normalization.

6. Max pooling using a $3 \times 3$ window.

7. Fully connected layer with 384 hidden units, and rectified linear activation.

8. Fully connected layer with 192, and rectified linear activation.

9. Output layer performing a linear projection to the output dimension (i.e. number of classes), returning logits.

The implementation can be found at our GitHub repository, referenced in the main paper.

An important detail in the GAM* experiments is the use of the agreement model predictions in the loss function of the classification model ($g_{ij}$ in Equation 1). In the graph experiments we used the agreement model predictions directly, with $g_{ij} \in [0, 1]$. However, if $g$ is trained to predict the probability of agreement, and the model is presented with roughly $50\%$ positive examples and $50\%$ negative examples, then whenever $g$ predicts values $< 0.5$, it believes that the two nodes should have different labels. Therefore, including such edges in the GAM regularization loss, albeit with a small contribution as predicted by $g$, still encourages agreement between nodes that $g$ believes should not agree. This does not pose a problem for the graph experiments, where the assumption is that the majority of the edges correspond to agreement (and $g$ reduces the contribution of those that do not). However, in the case of GAM*, we consider a fully-connected graph that contains all pairs of nodes. In most practical applications, this means that there are more edges between nodes that disagree than edges corresponding to agreement. Thus, a random batch of edges sampled for each of the regularization terms in Equation 1 will likely contain more disagreement edges, which all contribute a small loss that pulls the model parameters in the wrong direction. For this reason, in the GAM* experiments, we threshold the agreement model predictions at $0.5$. That is, we replace $g_{ij}$ in Equation 1 with ReLU($g_{ij} - 0.5$). In this manner, the only pairs of nodes that contribute to the GAM* regularization loss are those that the agreement model believes should have the same label.

## Footnotes

[3]`http://ufldl.stanford.edu/housenumbers/`