[Reviews · NeurIPS 2019]

Reviewer 1



The paper is clearly written and easy to understand. The paper addresses an important problem in semi-supervised learning where the graph edges and weights in a graph-based method might come from noisy sources. The proposed GAM method predicts the probability that pairs of nodes share the same label. The output of the agreement model is used to regularize the classification problem. To account for the limited labeled data, the classification and agreement models are jointly trained. Since the method can learn edges between datapoints, it addresses the issue common in cases when the graph is not provided, or the provided graph is incomplete in terms of having nodes that are disconnected from the graph. The experiments are thorough and explore the two settings of a given partial graph and having to learn the entire graph well across a variety of datasets. The key advantage of this method in constructing graphs comes from better predicting label agreement than using a distance metric between the nodes. Since the GAM uses the features of the nodes, which is generated by an encoder, the quality of the encoder is crucial to the performance of this method. Exploring how the quality of the encoder and noise in the features affects the quality of the GAM and the overall jointly trained model.

Reviewer 2



-------Originality The paper claims to propose a novel method in SSL that learns the graph instead of using a fixed graph. However, some closely related work sharing the same idea has been explored in [1, 4] and is, unfortunately, not mentioned in the paper. SNTG [1] is recent work on graph-based SSL, which also uses an auxiliary model to predict whether a pair of nodes is similar or not. The difference lies in the co-training part. [4] proposes a method based on dynamic infection processes to propagate labels. Please include [1,4] in the related work and add more discussions. -------Clarity The writing is really good. The paper is clear and easy to follow. -------Methodology 1. The introduction of agreement model incurs more network parameters. Is there any comparison with the baselines in terms of the number of network parameters? What is the performance of baselines with the same number of parameters as GAM? This seems to be a fairer comparison. 2. The convergence of the algorithm not guaranteed. Since the agreement model can make mistakes including the top confidant predictions, it may augment errors and propagate them into the classification model. How could the improvement be guaranteed at each iteration of the interaction in the co-training? Figure 5 in Appendix 5 also supports the concern. E.g., in Fig. 5(b) the test accuracy peaks at around iteration 15 and no longer improves after that (oscillates up and down). An extreme case that the augmented errors lead to diverged results may happen if the agreement model performs poor at the beginning and is learned slowly. See the sudden drop at about iteration 12 in Fig.5(a). It may not increase at the next iteration but gets worse. -------Experiments 1. What is the result of GCN_1024+VAT? It is not listed in Table 1. I noticed better results of GCN+VAT than those in this paper were reported in [2]. I was curious why VATENT failed as stated in line 291. Is it due to your implementation or the method itself? 2. In Sec. 4.2, a simple CNN is used for image classification. But this is not commonly used in the SSL literature. The paper only compared to VAT while several important baselines in SSL are missing. E.g., SNTG [1] and Fast-SWA [2] should be included in the related work and Table 3, which achieve much better performance than VAT. I recommend the authors use the standard stronger 13-layer CNN in the SSL literature and report the results, rather than omitting the closely related work [1, 3], especially SNTG which shares the same idea of learning a graph to improve the classification model. 3. How do you choose M, the number of most confident predictions? -----References--- [1] Smooth Neighbors on Teacher Graphs for Semi-supervised Learning, CVPR 2018. [2] Batch Virtual Adversarial Training for Graph Convolutional Networks, arxiv 1902.09192 [3] There Are Many Consistent Explanations of Unlabeled Data: Why You Should Average, ICLR 2019. [4] Semi-Supervised Learning with Competitive Infection Models, AISTATS 2018.

Reviewer 3



This paper proposes a new label-propagation method based on deep learning. Instead of propagation the labels directly from the graph, it learns another function which determines whether the edge is connected for same-label instances or not. The labels are then propagated according to the new determined connecting same-label edge. The learning classifier and propagating labels execute iteratively. Originality The limited novelty of the paper is mainly on the algorithm part. It first proposes to learn another function to determine whether the edge should connect two instances of the same label. Another is they use deep neural networks as base models for both tasks. The idea of using neural networks for semi-supervised learning has been explored for many years. Thus, the paper leads a small step to the deep-learning-based label propagation method. In the semi-supervised community, it has been discussed from long ago how to label the unlabeled data in a safe way instead of labeling them directly by classical semi-supervised methods. In this way, this paper can be counted as one in this trend and proposes a method to solve it, which simply learn whether two nodes should be joined or not. Besides, the paper studies a well-defined semi-supervised problem, without any new insights or problem setting. Quality One contribution of the paper is that they propose a training algorithm that resembles the co-training algorithm. However, the proposed method is not co-training. In co-training, two classifiers should both produce some data to feed to each other. In this proposal, only one classifier is generating data, and the other classifier is used only to update the graph. This is more like self-training instead of co-training. Other parts are technically sound and the experimental results are good. Clarity The paper is clearly written. Significant The paper may be of practical use since the empirical results shown are good. However, it only adds a small modification to existing algorithms, and replace all the classifiers by a deep learning one. In this way, the paper does not give too much new information (new problems, new insights, new assumptions) to the community and it has only limited significance. ===================================================== After the rebuttal, I will increase my score.

[Author Response · NeurIPS 2019]

We thank all reviewers for their thoughtful comments and suggestions. We address each review separately.

**Reviewer #1.** Regarding the encoder architecture impact, in Table 1, we vary the classification model making it increasingly more powerful, and demonstrate that our method produces improvements in all cases. For the agreement model encoder, we found that the results are not as sensitive to the encoder choice (e.g., for CIFAR10 switching from an MLP to a CNN did result in significant differences). We will include results for different agreement model architectures.

**Reviewer #5.** We thank R5 for the relevant references. We were not aware of them—especially the contemporaneous ones from ICLR and ICML 2019. SNTG infers a similarity graph between samples, but it does so in a significantly different way than GAM. Also, in contrast to SNTG, we propose an additional self-training component, and our method is applicable when a graph is provided, whereas SNTG (as published) is not designed to use information from a provided graph. We will include a thorough comparison to SNTG and Fast-SWA in the paper. We will also discuss the following:

- Parameters: Increasing the number of parameters of the baselines to match that of the respective GAMs results in worse performance for the baselines (e.g., in Table 1, $MLP_{256}$ has the same number of parameters as $MLP_{128}$+GAM, as we use an $MLP_{128}$ for the agreement model, but it performs worse than $MLP_{128}$+GAM and $MLP_{128}$). This is expected as GAMs provide a robust form of regularization for training high capacity models that tend to overfit otherwise.
- Convergence: Prior work [e.g., Blum and Mitchell 1998, Balcan et al. 2005] proves that co-training converges if: (i) the majority of the learners perform better than random guessing after the first iteration, and (ii) the mistakes they make are weakly dependent. Our experiments indicate that (i) is true in our case. (ii) is harder to verify due to the coupling between the models. However, our empirical evaluation shows that co-training converges successfully. Note that in Fig. 5, even the worse iterations are well above chance, so it should not diverge under these assumptions.
- Experiments: The missing numbers for $GCN_{1024}$+VAT are 83.4, 68.9, 79.5 on Cora, Citeseer, and Pubmed, while $GCN_{1024}$+VATENT obtains 32.5, 8.5, 18.0, which follow the same trend as our other results. For VATENT, we observed that on the graph datasets the entropy term becomes large and dominates the loss. Decreasing its weight makes the performance to converge to that of VAT. Our implementation works on CIFAR10 and SVHN, thus it seems unlikely to be the reason behind the poor results. Interestingly, [2] reports only GCN+VAT results and not GCN+VATENT.

Regarding comparisons with other methods, we will add the results reported in [2] to Table 1. Their best numbers are lower than our GCN+GAM. [4] tackles the same problem, but their evaluation is on random train/test splits rather than the commonly used Planetoid splits. We observe that the GCN paper reports much better results on random splits than [4], and we have demonstrated that GAM can be applied on top of GCN to improve it further. For completeness, we will report results on random splits and compare with [4]. To compare with SNTG and Fast-SWA, we plan to run the experiments with a 13-layer CNN suggested by R5 for the camera-ready. Note, however, that we do not necessarily see these approaches as competitors to GAM, but rather as additional regularizers that, similar to VAT, can be applied in conjunction with GAM to further improve generalization. To illustrate that GAM works for large networks too, here are results (obtained after the submission deadline) using the WideResnet of Oliver et al. 2018 on CIFAR10-4000: baseline 79.69%, +$\Pi$-Model 83.63%, +Mean teacher 84.13%, +VATENT 86.87%, +GAM* 87.42%.

**Reviewer #6.** R6 suggests a discussion on the challenges in simply replacing classification models in label propagation with deep learning models. We address this through an example from our paper, and then explain how this example is more broadly applicable. Replacing classification models in label propagation with deep learning models is exactly what Neural Graph Machines (NGMs) do (described in Section 2): an NGM is a label propagation model complemented by a deep learning classifier operating on the node features. Setting the regularization coefficients to 0 makes it a pure deep learning model, while increasing their values brings it closer to label propagation. When the graph is noisy, the regularization coefficients need to be small (otherwise the regularization forces connected nodes from different classes to incorrectly have the same label), thereby reducing the effect of the graph on the model. However, with such minimal regularization the model tends to overfit to the few available labeled examples. Our approach combines deep learning with label propagation in a manner that allows us to handle noisy graphs in a robust fashion. Note that other methods besides NGM also suffer from this problem (e.g., GCN, Planetoid)—see *Robustness* section. Our experiments show how GAMs are able to learn in a much more robust manner.

Novelty: The novelty of our algorithm is the interaction between the agreement and classification models, which allows it to benefit from both label propagation and deep learning even when dealing with noisy graphs (where most label propagation algorithms fail), or no graphs at all. It is surprising and interesting that even though the two models learn using the same features and same data, their interplay can produce such large increases in accuracy on a wide variety of base networks (MLP, CNN, Resnet, GCN, and GAT), suggesting they learn complementary information.

Co-Training: We argue that our proposed training algorithm does indeed fit in the co-training framework. While the original paper [Blum and Mitchell, 1998] proposed co-training in the setting described by R6, the same authors subseqeuently proposed co-training settings where some classifiers predict label distributions and others predict coupling constraints over these distributions (like in our setting). Perhaps the most notable and influential example of this is the Never-Ending Language Learning (NELL) system [Mitchell 2015, 2018].

[Meta-Review · NeurIPS 2019]

This paper proposed a novel graph learning method for graph-based semi-supervised learning. Besides the model of the classifier (the classification model in the paper), another model of the graph is considered (the agreement model in the paper), and the contribution to the loss of each edge is determined by the model of the graph. Although there are still concerns about the novelty in the end, we all agree that the proposed method is simple, well-explained and can still achieve good performance. This may have impacts to practitioners using semi-supervised learning in their projects, and as a result, I recommend an acceptance. Note that an important direction of related work is missing, namely, non-parametric graph learning methods for graph-based semi-supervised learning, see https://papers.nips.cc/paper/4801-forging-the-graphs-a-low-rank-and-positive-semidefinite-graph-learning-approach from NeurIPS 2012 and references therein (I am not a coauthor of any of them). Please survey this direction and include your survey in the final version.